# CD44: A New Prognostic Marker in Colorectal Cancer?

**DOI:** 10.3390/cancers16081569

**Published:** 2024-04-19

**Authors:** Pina Ziranu, Andrea Pretta, Valentina Aimola, Flaviana Cau, Stefano Mariani, Alessandra Pia D’Agata, Claudia Codipietro, Daiana Rizzo, Veronica Dell’Utri, Giorgia Sanna, Giusy Moledda, Andrea Cadoni, Eleonora Lai, Marco Puzzoni, Valeria Pusceddu, Massimo Castagnola, Mario Scartozzi, Gavino Faa

**Affiliations:** 1Medical Oncology Unit, University Hospital and University of Cagliari, SS 554 km 4500 Bivio per Sestu, Monserrato, 09042 Cagliari, Italy; an.pretta@gmail.com (A.P.); mariani.step@gmail.com (S.M.); alessandrapiadagata@gmail.com (A.P.D.); claudiacodipietro96@gmail.com (C.C.); daiana.rizzo94@gmail.com (D.R.); veronicadellutri@outlook.it (V.D.); giorgia.sanna98@hotmail.it (G.S.); giusymo.17@gmail.com (G.M.); cadoni.andrea@outlook.it (A.C.); ele.lai87@gmail.com (E.L.); marcopuzzoni@gmail.com (M.P.); valeria.pusce@gmail.com (V.P.); marioscartozzi@gmail.com (M.S.); 2Division of Pathology, Department of Medical Sciences and Public Health, AOU Cagliari, University of Cagliari, 09124 Cagliari, Italy; vale.aimola@gmail.com (V.A.); flacau@tiscali.it (F.C.); 3Proteomics Laboratory, Centro Europeo di Ricerca sul Cervello, IRCCS Fondazione Santa Lucia, 00013 Rome, Italy; massimo.castagnola@icrm.cnr.it; 4Department of Medical Sciences and Public Health, AOU Cagliari, University of Cagliari, 09124 Cagliari, Italy; gavinofaa@gmail.com; 5Department of Biology, College of Science and Technology, Temple University, Philadelphia, PA 19122, USA

**Keywords:** colorectal cancer, CD44, cancer stem cells, prognostic marker, predictive marker, target therapies

## Abstract

**Simple Summary:**

CD44 is a crucial factor in colorectal cancer, with specific isoforms demonstrating their significance in the development, progression, metastasis, and resistance to therapy. Given the clinical and pathological impact of CD44, it represents a promising molecular target for cancer therapy. In this review, we aim to highlight the predictive and prognostic significance of CD44 in various cancer types, with a particular focus on colorectal cancer. Moreover, we evaluate current therapeutic interventions that target CD44 or reduce its expression, thereby highlighting its potential as an effective therapeutic strategy.

**Abstract:**

Cluster of differentiation 44 (CD44) is a non-kinase cell surface glycoprotein. It is overexpressed in several cell types, including cancer stem cells (CSCs). Cells overexpressing CD44 exhibit several CSC traits, such as self-renewal, epithelial–mesenchymal transition (EMT) capability, and resistance to chemo- and radiotherapy. The role of CD44 in maintaining stemness and the CSC function in tumor progression is accomplished by binding to its main ligand, hyaluronan (HA). The HA-CD44 complex activates several signaling pathways that lead to cell proliferation, adhesion, migration, and invasion. The CD44 gene regularly undergoes alternative splicing, resulting in the standard (CD44s) and variant (CD44v) isoforms. The different functional roles of CD44s and specific CD44v isoforms still need to be fully understood. The clinicopathological impact of CD44 and its isoforms in promoting tumorigenesis suggests that CD44 could be a molecular target for cancer therapy. Furthermore, the recent association observed between CD44 and KRAS-dependent carcinomas and the potential correlations between CD44 and tumor mutational burden (TMB) and microsatellite instability (MSI) open new research scenarios for developing new strategies in cancer treatment. This review summarises current research regarding the different CD44 isoform structures, their roles, and functions in supporting tumorigenesis and discusses its therapeutic implications.

## 1. Introduction

Cancer development is a complex process that involves genetic abnormalities and genome instability, which enables it to invade and metastasize [1,2]. However, genetic alterations alone cannot guarantee tumor growth. The selection of a malignant phenotype arises from the accumulation of many metastasis-promoting activities involved in a multi-step process [3,4,5]. Invasion is the first step in the metastasis process, which involves cancer cells penetrating the basement membrane and moving through the extracellular matrix (ECM) into the surrounding tissue [6,7]. Tumor cell migration is driven by various factors in the tumor microenvironment, such as hypoxia, chemoattractants, ECM stiffness, and nutrient depletion. Adhesion molecules play a crucial role in this process, and cancer cells use the same adhesive functions as normal cells to carry out their physiological activities [8].

Epithelial–mesenchymal transition (EMT) is a biological process used during embryogenesis and adult epithelial tissue healing. However, it can be hijacked by cancer cells to acquire malignant features [9]. To transform from epithelial cells to mesenchymal cells, specific transcription factors alter the gene-level expression of surface markers. This results in reduced epithelial cell markers and increased mesenchymal cell markers, leading to a change in cell phenotype [10]. The cells generated through this process can self-renew and are called cancer stem cells (CSCs) [11]. Various studies demonstrated that CSCs express the surface marker CD44 (Cluster of Differentiation 44). CD44 is a transmembrane, non-kinase, single-chain glycoprotein that was initially identified in lymphocytes [12] and later found in various human tissues. Further research by Stamenkovic and colleagues showed that this antigen is also expressed in numerous solid tumor cell lines [13]. Subsequently, Günthert and colleagues discovered that an isoform of CD44 could modify the aggressiveness of a tumor by giving it metastatic properties when inserted into its genetic sequence [14].

CD44 has been extensively studied in recent years, and clinical and preclinical studies demonstrated its role as a marker of progression and resistance to therapy in several types of cancer. Specifically, CD44 appears to play a crucial role in colorectal cancer (CRC), where specific isoforms have been shown to play a central role in carcinogenesis, progression, metastasis, and resistance to therapy. Additionally, the clinicopathological impact of CD44 suggests that it may be a molecular target for cancer therapy. This review aims to discuss the prognostic and predictive importance of CD44 in cancer diseases, especially in CRC. It will also evaluate current therapeutic strategies that target CD44 or reduce its expression.

### 1.1. Structure of CD44

The CD44 protein is encoded by a single gene on chromosome 11p13, which is expressed ubiquitously throughout the body [15]. This gene contains approximately 20 exons and produces a protein divided into three regions: the extracellular domain, the transmembrane region, and the cytoplasmic domain [16].

The smallest and most common isoform of CD44, CD44 standard (CD44s), is present in all isoforms and is encoded by the first five exons (exons 1–5) and the last five exons (exons 16–20) [17]. The first five exons encode the amino-terminal sequence, which is highly conserved. This sequence contains the recognition of hyaluronan (HA), an abundant component of ECM expressed by stromal and tumor cells, and the primary ligand of CD44 [18,19]. The region proximal to the membrane of the extracellular domain is less conserved, and the ten central exons can give rise to variants (CD44v) through an alternative splicing process, commonly known as “v1–v10”. These variants can be inserted in different exon 5 and 16 combinations, increasing protein variability [19,20].

CD44’s extracellular domain can bind to various ligands, including HA, osteopontin (OPN) [21], chondroitin [22], collagen [23], fibronectin [24], and serglycin/sulfated proteoglycan [25]. Among these, HA is the most specific ligand for CD44 activation. All forms of CD44 have an HA-binding domain located in the N-terminal region of the extracellular domain. When HA binds to CD44, it causes changes in the protein’s shape that promote the binding of adaptor molecules to the intracellular cytoplasmic tail of CD44 (Figure 1). This leads to cell signaling that enhances cell adhesion, migration, and proliferation [26].

The transmembrane domain of CD44 is responsible for interacting with cofactors, adaptor proteins, and protein tyrosine kinases to regulate CD44 activation [27].

The cytoplasmic domain (CD44ICD) is essential in regulating signal transduction, mediating intracellular signaling, and influencing various cellular processes [28]. Although the CD44ICD lacks enzymatic activity, it possesses specific structural motifs facilitating interactions with cytoplasmic effectors and regulating cell-trafficking machinery, signal transduction pathways, the transcriptome, and crucial metabolic pathways [29].

HA-CD44 binding triggers signaling events within cells by affecting the interaction between the CD44ICD and downstream signaling molecules such as the actin cytoskeleton [30], ezrin, radixin, and moesin (ERM) proteins [31], as well as ankyrin [32], regulating cell trafficking and cytoskeletal organization. In particular, the interaction between CD44 and ERM proteins promotes cell growth and migration. However, this interaction is countered by the binding of merlin, a protein distantly related to ERM family proteins, thus leading to the dissociation of CD44 from the cytoskeleton [33,34]. This competition between ERM proteins and merlin for binding to CD44 regulates the effects of CD44 on cell growth and migration. This mechanism is crucial in contact inhibition of cells [35]. Furthermore, the interaction of CD44 with IQGAP1, an actin-binding protein that regulates cell–cell and cell–matrix adhesion, was also observed in standard and cancer cells. IQGAP1 enhances the activities of both RhoA and Rac1 GTPases by stabilizing their GTP-bound forms, promoting Rho-dependent F-actin stress fibers formation, but does not influence ERK1/2 activation [36,37]. Ankyrin was the first intracellular protein to partner with CD44ICD. It connects CD44ICD to the cytoskeleton, resulting in the formation of caveolae on the cell membrane. These contain specific proteins and lipids and initiate signaling cascades that regulate many cellular processes, including cell proliferation, adhesion, migration, and communication. Examples of the signaling cascades include Ca^2+^ mobilization and release, PI3K-AKT signaling, and RhoA-GTPase signaling [38].

In addition, CD44ICD has the ability to interact with Rho-family GTPases and members of the Src family of non-receptor tyrosine kinases [39], activating PI3 kinase/Akt signaling [38]. These interactions regulate pathways related to cell growth, survival, differentiation, stemness, and therapeutic resistance [29].

Post-translational modifications, such as N-linked and O-linked glycosylation and glycosamino-glycanisation, can add heparan sulfate or chondroitin sulfate to CD44, which increases the variability of this protein and its isoforms [16].

### 1.2. Functional Significance of CD44 Isoforms in Cancer Cells

CD44 is a protein expressed in most vertebrate cells, while CD44v isoforms are found only in specific cells, particularly in aggressive tumors [40]. Different isoforms of CD44 have distinct functions. They have unique properties, such as interacting with the microenvironment, acting as co-receptors, and activating additional signaling pathways [41]. CD44 interacts with several ligands, including HA, chondroitin, collagen, laminin, and fibronectin, which are responsible for their ability to promote tumor progression and increase aggressiveness [42,43]. By interacting with HA and other ECM components, CSCs can perceive and integrate information from the tumor microenvironment, mediate signal transduction, and maintain stemness characteristics [44,45]. CD44v isoforms such as CD44v6, CD44v3, and CD44v2 are markers of progression and resistance to therapy in various cancers [46,47,48,49]. CD44 and its isoforms have potential prognostic and predictive roles in cancer treatment response, making them potential targets for anticancer therapies.

Certain CD44 variants’ functional roles in tumor growth differ from those of standard CD44s. As a result, it is imperative to ascertain the importance of each CD44 variant and its correlation to tumor progression. Different variants of CD44 may possess both overlapping and distinct functions. Many studies have attempted to define the role of different CD44 isoforms in CRC carcinogenesis and progression.

CD44v8-10 proteins are located in the stem cell niche of a normal human colon. They are frequently overexpressed in early colon adenomas. When colon tumors form, CD44v8-10 becomes overexpressed, leading to the overpopulation of CD44v8-10+ cancer stem cells during CRC progression [50,51,52]. Furthermore, CD44v8-10 protects gastrointestinal cancer cells against reactive oxygen species by directly interacting with a glutamate-cystine transporter [53]. Knock-in mice expressing CD44v4-10 promote adenoma initiation in Apc (Min/+) mice but not in CD44 knock-in mice [54].

The variants CD44v6 and v7/8 are up-regulated by hypoxia-inducible factor (HIF) under hypoxic conditions [55]. CD44v6 and CD44v10 mRNA is detected in CRC patients [56]. Cytokines such as hepatocyte growth factor (HGF), OPN, and stromal-derived factor 1α (SDF-1) up-regulate CD44v6 expression in CSC, which activates the Wnt/β-catenin pathway responsible for the promotion of cancer cell migration and metastasis in CRC [57]. Overexpressing CD44v6 increases cell viability, clonogenicity, autophagy flux, EMT, and phosphorylation of AKT and ERKs in CRC cells in the presence of chemotherapy drugs [58].

### 1.3. CD44 Activation and CSC Stemness

CSCs are long-living cells that play a critical role in cancer progression. They contribute to tumor growth and may be responsible for the ineffectiveness of some cancer treatments [59,60,61]. CSCs exist in a unique environment that includes immune cells, micro-vesicles, and cytokines, which promote self-regeneration and metastasis while suppressing the immune system [62]. CSCs maintain stemness features by interacting with the ECM through specific membrane receptors [63]. This interaction occurs in the CSC niche’s unique microenvironment [28]. Research has revealed that information encoded in the ECM can direct the differentiation of stem cells towards a specific cell type [64,65,66]. HA, a glycosaminoglycan, is a crucial component of stem cell niches that binds to CD44 receptors on CSCs to encourage cell growth, wound resolution, and cell motility [11,67,68,69,70,71]. Likewise, to normal stem cell niches, HA is a major constituent of the CSC niche [72]. Its interactions with its main receptor, CD44, play a significant role in various aspects of tumorigenesis and cancer progression. HA binds to CD44 and activates multiple cell surface receptors, modulating oncogenic signaling pathways and regulating cell migration. This interaction also leads to drug resistance. Given their involvement in multiple cellular functions, CD44 and HA are linked with regulating CSC properties such as self-renewal and tumor initiation [43,73].

During embryogenesis and tissue healing, its expression is high [74]. Increased HA before cell division enables cancer cells to detach from the ECM and lose their binding to neighboring cells [75,76,77]. Irregular synthesis or degradation of HA may lead to abnormal cellular processes, such as proliferation, cell transfer, and metastasis [78,79]. HA production begins with two molecules, which HA synthases use to create polymers of different sizes [80,81]. Hyaluronidases can break down high-molecular-weight HA polymers (HMWHA) into smaller but still biologically active molecules (LMWHA) [82]. CD44 binding to HMWHA is linked to cell proliferation and tissue growth, whereas LMWHA regulates neovascularization phenomena and is involved in the induction of pro-inflammatory processes [83,84].

Upon the binding of HA to the extracellular domain of CD44, conformational changes occur, leading to CD44 activation and the recruitment of various cytoplasmic and membrane-bound proteins (adaptor molecules) (Figure 1). The resulting downstream cell survival, growth, and tumor progression pathways are triggered [67]. The HA ligand induces a signal cascade by stimulating the CD44ICD, which results in the recruitment of numerous proteins and a series of cell signaling events. This includes the activation of proteins such as ankyrin, merlin, and ERM protein, which mediate actin polymerization through ERM proteins, facilitating cytoskeleton rearrangements for tumor invasion and migration [85]. Additionally, CD44ICD has the ability to coordinate signaling responses because many intracellular signaling molecules interact with it, including Rho-family GTPases and members of the Src family of non-receptor tyrosine kinases. These molecules, in turn, activate PI3 kinase/Akt signaling [29,38,39].

The binding of HA to its receptor CD44 is mediated through the activation of the Rho GTPase pathway, which subsequently activates PI3K and the serine/threonine kinase (Akt) cascade and RAS-RAF-MAPK signaling pathways. This activation results in the phosphorylation of substrates responsible for various cellular processes, including replication, growth, and mobility [86]. CD44 isoforms can interact with several cell surface receptors, such as c-Met, vascular endothelial growth factor receptor-2 (VEGFR-2), platelet-derived growth factor receptor (PDGFR), and epidermal growth factor receptor (EGFR), which are implicated in the progression of numerous tumors [82,87,88]. The binding of CD44 with other components of the ECM is made easier by post-translational modifications of CD44, such as the addition of chondroitin sulfate and heparan sulfate to the amino acid sequences of the variable region. This allows CD44 to interact with collagen, fibronectin, and laminin. Additionally, sulfation of the sugar side chains of CD44 enables it to bind to fibrin [22,89,90].

In summary, CD44 binding with different components of the ECM can regulate and induce cancer cell growth and metastasis.

The binding of HA-CD44 triggers the activation of downstream signaling molecules, such as the actin cytoskeleton, ezrin, radixin, and moesin (ERM) proteins, and ankyrin, regulating cellular trafficking and cytoskeletal organization. In addition, HA-CD44 can interact with Rho family GTPases and Src family members of non-receptor tyrosine kinases, activating PI3-kinase/Akt signaling. These interactions regulate cell growth, survival, differentiation, stemness, and therapeutic resistance pathways.

## 2. Prognostic Role

CD44 plays a crucial role in preserving the stemness and function of CSCs during tumor progression. As a result, it can be a useful prognostic marker. However, its prognostic value is still debated (see Table 1).

Numerous studies confirmed the crucial role of high CD44 expression in the development of cancer, tumor growth, differentiation, and metastasis in CRC [91,92,93,94]. In a study conducted by Weber et al., it was observed that knocking out the CD44 gene prevented tumor metastasis, though the tumor continued to exist in mice [94]. Research conducted in vitro has shown that a single CRC cell can produce highly heterogeneous CRCs if it expresses CD44 [95,96]. Some researchers reported that the expression level of CD44 was higher in high-grade CRCs when compared to low-grade tumors, and this overexpression was associated with reduced patient survival [97,98]. In various types of malignancies, such as lymphomas, gastric, and cervical carcinomas, higher expression of CD44 has been recorded in advanced tumors, which might be related to poor prognosis [99,100]. Furthermore, in our recent study, high CD44 expression was significantly associated with higher proliferative activity of CRC and poor prognosis. CD44 overexpression was also associated with clinically poor prognostic features: older age, inoperable disease, stage IV at diagnosis, mutated BRAF, and high-grade tumor [101,102]. Conversely, in a previous study, a statistically significant correlation was found between positive CD44 expression and left-sided tumors in an Egyptian population [103], which tend to be associated with a better prognosis than right-sided CRC localization [104,105].

Recent research suggests that the absence of CD44 is associated with a worse prognosis [106,107,108]. In one study, Hong et al. examined 162 patients and found that low CD44 expression was linked to an increased risk of tumor recurrence and shorter disease-free survival (DFS) [106]. Similarly, Qu et al. reported that low CD44 expression was significantly associated with lower overall survival (OS) and DFS in stage II and III CRC patients [107]. Another study by Lugli et al. analyzed 1420 cases and found that the loss of membranous CD44 expression was related to higher tumor stage and lymph node involvement, an infiltrative growth pattern, and vascular invasion. Specifically, the loss of CD44 was associated with an increased likelihood of local cancer recurrence [108,109].

This inconsistency in the results could be due to the alternative splicing of CD44 pre-RNA. Alternative splicing is a common mechanism that generates multiple mRNA isoforms from a single gene. CD44 pre-RNA undergoes alternative splicing to produce various CD44 isoforms that have different extracellular domains and cytoplasmic tails. These isoforms are involved in different cellular processes, including tumor progression and metastasis [110,111,112]. Recent studies have revealed that DNA methylation plays a crucial role in regulating CD44 alternative splicing. DNA methylation can modulate the recruitment of RNA polymerase II and chromatin factors to the CD44 gene. A loss of intragenic DNA methylation in CRC cells has been found to increase CD44 variant exon skipping, leading to a partial epithelial to mesenchymal transition [110].

The exact role of different CD44 variants in cancer is still not well understood. While they overlap, they also have distinct roles. CD44v isoforms contain additional binding sites that enable CD44 to interact with molecules in the microenvironment. Some studies suggest that certain CD44 variants may play a crucial role in the development of metastasis in CRC. Yan et al. discovered that CD44, CD44v3, and CD44v6 are expressed heterogeneously in CRC. Their clinical study revealed that the prognostic value of CD44 and its splice variants is not always unanimous in CRC. Patients who lacked CD44, or those who had an expression of CD44v3 and v6, had reduced progression-free survival (PFS) [113].

**Table 1 cancers-16-01569-t001:** Prognostic role of cluster of differentiation 44 (CD44).

Study	Country	N. of pts	Year	Cancer Types	CD44 Expression	Results
Bendardaf R et al. [92]	Finland	95	2005	Colorectal	High CD44v6	Greater T
Liu JL et al. [93]	China	62	2005	Colorectal, gastric, breast, lung cancer	HighCD44s and CD44v6	Greater N and G
Weber et al. [94]	USA	Preclinical	2002	Sarcoma	Absence of CD44	No metastasis formation
Du L et al. [95]	China	60	2008	Colorectal	CD44+	In vitro: generate xenograft tumor
Vermeulen L et al. [96]	Netherland Italy	Preclinical	2008	Colorectal	Single-cell-cloned CSC CD44+	In vitro: generate an adenocarcinoma on xenotransplantation
Ropponen KM et al. [98]	Finland	194	1998	Colorectal	HighCD44v3 and CD44v6	Lower RFS
Carr NJ et al. [99]	England	299	2002	Appendiceal Colorectal	CD44s +	Higher in colorectal
Zhao LH et al. [100]	China	187	2015	Colorectal	High CD44 and CD44v6	Greater TNM and poorly differentiated histology
Ziranu P et al. [101]	Italy	65	2023	Colorectal	CD44 3+	Lower mOS and clinically poor prognostic features
Holan NS et al. [103]	Egypt	71	2022	Colorectal	CD44+	Higher in left-sided colon cancer
Hong I et al. [106]	Korea	162	2015	Colorectal	Low CD44	Increased tumor recurrence and lower DFS
Qu et al. [107]	China	223	2017	Colorectal (stage II–III)	Low CD44	Lower mDFS and mOS
Lugli A et al. [108]	Switzerland	1420	2010	Colorectal	Low CD44	Higher TN
Yan B et al. [113]	China	148	2020	Colorectal	CD44− CD44v+ CD44v6+	Lower PFS
Ozawa M et al. [48]	Japan	77	2014	Colorectal	CD44v2+	Greater TNM Worse prognosis
Vizioso F et al. [114]	Spain	105	2001	Colorectal	CD44v5 and CD44v6	Lower PFS and OS
Zalewski B [115]	Poland	114	2004	Colorectal	CD44v5 and CD44v6	No impact on prognosis
Yamaguchi A et al. [116]	Japan	71	1998	Colorectal	CD44v8-10	Greater N and liver metastases
Nihei Z et al. [117]	Japan	42	1996	Colorectal	CD44v6	Lower mOS
Bendardaf R et al. [118]	Finland	57	2004	Colorectal	High CD44v6	Increased treatment response

Legend: pts = patients; T = primary tumor stage; N = locoregional lymph node stage; G = tumor grading; RFS = relapse-free survival; OS = overall survival; PFS = progression-free survival.

The expression of CD44v2 is associated with a poor prognosis [48], while the expression of CD44v5 and CD44v6 is believed to be linked to a shorter relapse-free survival [114], although there is some controversy surrounding this [115]. CD44v8-10 is associated with lymphatic and venous invasion as well as liver metastasis [116]. CD44v6, on the other hand, has shown promising potential as a biomarker since several studies have revealed that its overexpression is linked to lymph node and distant metastases, treatment response, and tumor-associated mortality [100,117,118].

In CRC patients, CD44v6 and CD44v10 mRNA were found, and CD44v6 was expressed in colorectal CSC. CSC required CD44v6 to migrate and generate metastatic tumors. Cytokines like HGF, OPN, and SDF-1 secreted in the tumor microenvironment increased CD44v6 expression in CSCs and activated the Wnt/β-catenin pathway, which promoted migration and metastasis [119].

There is an ongoing debate among experts regarding the significant association of CD44 with a poor prognosis in various cancers. However, more reliable analyses suggest that increased CD44 expression is linked to aggressive clinical and histopathological features, such as advanced clinical stage, higher histologic grade, and tumor stage. This further suggests a shorter survival time for the patient, indicating a poorer prognosis.

The heterogeneity of methods and scores used in various studies contributes to the conflicting data on the prognostic role of CD44. Our recent study [101,102] utilized an established expression score scale for HER2 in breast cancer to assess CD44 expression. We found that the intensity of staining, rather than the percentage of positive cells, was the significant factor. Therefore, we chose a scoring system modeled on the HER2/neu scheme, which correlated positively with the patient’s prognosis and clinicopathological features. High expression, defined as intense membrane staining in at least 10% of tumor cells (score 3+), could aid in identifying patients with poor prognosis in mCRC.

## 3. Predictive Role

Understanding the molecular mechanisms of treatment resistance in CRC is essential for choosing the most effective therapies [120,121,122,123,124]. The CD44 glycoprotein is a surface marker of CSCs that activates signaling pathways, promoting cancer cell metastasis, adhesion, migration, and drug resistance [119]. Although CD44 lacks kinase activity, it can activate protein kinases through several mechanisms. CD44 sequesters growth factors at the cell surface and stabilizes tyrosine kinase receptor complexes as a co-receptor (as HGF, VEGFR-2), modulating downstream signaling pathways [30]. Furthermore, CD44 binding to HA triggers signaling cascades [11,67]. The activation of Src family kinases, as well as various GTPases (e.g., RhoA, Ras, and Rac1), drives intracellular signaling pathways, such as the Ras-MAPK and PI3K/Akt pathways, that promote tumor cell-specific processes, including chemoresistance [82,87,88]. A significant resistance mechanism mediated by HA-CD44 involves the formation of a complex between Nanog (an embryonic stem cell transcription factor) and the “signal transducer and activator of transcription protein 3” (Stat-3). This complex up-regulates the multidrug resistance (*MDR1*) gene expression, increasing the drug pump efflux induced by the cytoskeletal protein ankyrin [125].

Chemoresistance seems to be associated with the CSCs, expressing specific CD44v isoforms [126] (see Table 2).

For example, in human head and neck squamous cell carcinoma, Nanog activation by the HA-CD44v3 promotes the expression of pluripotent stem cell markers, including Nanog-Sox2-Oct4 complexes. These complexes promote CSC stemness, a harmful factor for chemosensitivity [127].

Recently, chemo-sensitive analysis was conducted on human CRC cell lines (COLO 201) divided into two populations based on CD44 expression: CD44 positive (CD44+) and CD44 negative (CD44-). COLO 201 CD44+ cell lines showed more stemness properties and lower sensitivity to 5-fluorouracil in vitro compared to COLO 201 CD44-negative cell lines [128].

Toden et al. demonstrated, through in vitro data, that the CD44v6 CSC cell line showed increased resistance to 5-Fluorouracil and Oxaliplatin, thus supporting the hypothesis that CD44v6, frequently overexpressed in CSCs in advanced CRC, confers higher stemness and increased resistance to chemotherapeutic drugs [96]. The assessment of CD44v6-positive circulating tumor cells (CTCs) at baseline is also associated with treatment failure, reinforcing that CD44v6 expression may reflect a biomarker of intrinsic resistance to treatment [129].

Furthermore, the protein CD44v6 has been identified as a co-receptor of vascular endothelial growth factor (VEGF) [130]. CD44v6 and CD44v9 positive cells exhibit an antiapoptotic effect and can block Fas-mediated apoptosis. The activity of immune checkpoint inhibitors might be reduced by interfering with Fas signaling through CD44v isoforms [131]. Consequently, detecting CD44v6- or CD44v9-positive CTCs could be a valuable tool in monitoring the development of drug resistance during anti-angiogenic therapy or immunotherapy.

The usefulness of CD44 in evaluating the therapeutic responses of targeted therapy has been analyzed in glioblastoma multiform (GMB). In vivo and In vitro, CD44 reduces the antitumor effect of bevacizumab, resulting in much more highly invasive tumors [132].

Moreover, Chen S. et al. found a significant correlation between CD44 levels and immune-infiltrating cells such as T cells, B cells, NK cells, and macrophages in various cancer types. CD44 also showed a connection with tumor mutational burden (TMB) and MSI, indicating its potential as an emerging biomarker for predicting responses to immunotherapy [133].

CD44 could act as a KRAS regulator in promoting the development, progression, and stemness of CRC. KRAS mutations are found in 40–50% of CRCs and are a critical biomarker for metastatic CRC [134]. The mutational status of KRAS can predict the efficacy of biological drugs, such as anti-EGFR antibodies [135,136]. KRAS activation plays an essential role in the cell signal transduction pathways, including PI3K/Akt and RAS-RAF-MAPK signaling pathways, which are involved in cell proliferation [137]. Recent data showed that inhibiting KRAS activation is a potential strategy for treating CRC [138,139,140,141].

**Table 2 cancers-16-01569-t002:** Predictive role of cluster of differentiation 44 (CD44).

Study	N. of Patients	Year	Cancer Cell	CD44 Expression	Results
Bourguignon L.Y.W. et al. [127]	Preclinical	2012	Human HNSCC-derived HSC-3 cells	CD44v3	HA-induced CD44v3 interaction with Oct4-Sox2-Nanog signaling promotes self-renewal, clonal formation, and cisplatin resistance
Okuyama H et al. [128]	Preclinical	2020	CRC cell lines	NA	CD44+ cell lines showed more stemness properties and lower sensitivity to 5-fluorouracil
Toden et al. [126]	Preclinical	2019	CRC cell lines	CD44v6	CD44v6 CSC cell line showed increased resistance to 5-Fluorouracil and Oxaliplatin
Nicolazzo C et al. [129]	40 pts	2020	mCRC	CD44v6	CD44v6-positive CTC predict treatment failure
Tremmel M et al. [130]	Preclinical	2009	Variety cancer cell lines	CD44v6	CD44v6 has been identified as a co-receptor of VEGF
Mielgo A. et al. [131]	Preclinical	2006	Jurkat cells and plasmacytoma cell lines	CD44v6 CD44v9	CD44v6 and CD44v9 exhibit an antiapoptotic effect and can block Fas-mediated apoptosis
Nishikawa M et al. [132]	Preclinical	2021	GBM	CD44	Bev showed no antitumor effects in mice transplanted with CD44-overexpressing GSCs
Chen S et al. [133]	Preclinical	2023	Pan-cancer	CD44	CD44 expression was significantly associated with TMB and MSI
Zhao Y et al. [142]	Preclinical	2021	GBM	CD44	KRAS/ERK pathway regulates CD44 overexpression in response to radiation by downregulating micro-RNA expression
Zhao P et al. [143]	Preclinical	2013	Lung cancer	CD44	CD44 mediates KRAS-dependent MAPK activation and cell proliferation
Ribeiro KB et al. [144]	58 pts	2016	mCRC	CD44	CD44 expression and KRAS mutation are correlated

Legend: HNSCC = head and neck squamous cell carcinoma; CRC = colorectal cancer, mCRC = metastatic colorectal cancer; CTC = circulating tumor cells; VEGF = vascular endothelial growth factor; GBM = glioblastoma; Bev = bevacizumab; GSCs = glioma stem-like cells; pts = patients.

In a study involving GMB tumor cells, it was demonstrated that the KRAS/ERK signaling pathway regulates the overexpression of CD44 in response to radiation by downregulating micro-RNA expression in GBM cells [142]. In a lung cancer setting, a mouse model of KRAS-induced lung adenocarcinoma and KRAS and CD44 expression were evaluated to determine the in vivo role of CD44 related to KRAS. The findings revealed that CD44 expression is up-regulated in KRAS-induced lung adenocarcinomas and that CD44 mediates KRAS-dependent MAPK activation and cell proliferation [143]. Moreover, in a clinical trial, Ribeiro et al. found that CD44 expression and KRAS mutation in metastatic CRC are correlated. Patients with mutated KRAS and positive immunoreactivity for CD44 had a worse prognosis. The study recommended closer monitoring during and after therapy for this subgroup of patients. With all these data taken together, CD44 could represent a potential therapeutic target for treating KRAS-dependent carcinomas [144].

Therefore, the role of CD44 in regulating the properties of CSCs and tumor cell signaling pathways makes it a critical factor in developing drug resistance. The identification of CD44 as a predictor of drug resistance suggests that targeting CD44 could be an effective strategy to improve response to therapies.

## 4. CD44 as a Possible Therapeutic Target

CD44 expression plays a significant role in tumorigenesis, progression, and chemoresistance. Therefore, inhibiting CD44 could be a potential therapeutic target for cancer treatment. There are several ways to target CD44, including neutralizing antibodies, antibodies, peptides, pharmacological and natural inhibitors, HA-modified nanocarriers, small interfering RNAs (siRNAs), and CAR T cell therapy [145]. These therapies are all currently being evaluated in preclinical and clinical settings in different types of cancers.

In the contest of CD44-targeted antibodies, monoclonal antibody (mAb)-modified Doxil against CD44 showed a significant improvement in cellular uptake and higher doxorubicin concentration inside tumor cells than Doxil in CD44-expressing murine CRC cells [146]. Another antibody, the humanized mAb RG7356, induced cytotoxic effects in chronic lymphocytic leukemia cells and had a significant immune-stimulatory effect. RG7356-binding CD44+ tumor cells stimulated the secretion of chemoattractants and facilitated the recruitment of immune cells, such as macrophages, leading to antibody-dependent cellular phagocytosis of cancer cells by macrophages [147,148]. A phase I clinical trial for RG7356 showed limited results with modest therapeutic efficacy in heavily pre-treated metastatic solid tumors, including CRC [149]. A study by Birzele et al. showed that CD44 expression is associated with HA production and can predict the response to treatment with RG7356 in tumor xenograft models [150].

Studies conducted on the PC3 prostate cancer cell line showed that F77, a prostate cancer-specific mAb carried by CD44, induced apoptosis in a CD44-dependent manner [151]. Encapsulated glycosylated paclitaxel liposomes (gPTX-L) conjugated with anti-CD44 antibodies enhance cytotoxicity efficiently in vitro and in vivo in human ovarian overexpressed CD44 cancer cell lines [152].

Interestingly, the potential of synthetic peptides to selectively bind to CD44 has been recently reported. Peptides have robust physicochemical properties, making them superior to antibodies for diagnostic and therapeutic purposes. P7 (FNLPLPSRPLLR) demonstrated the highest affinity and specificity for CD44 in breast CSCs among seven different peptides discovered [153]. PDPP, a polyvalent-directed peptide polymer, recognized breast cancer stem cells using combinational peptides P6 and P7, displaying a higher affinity and inhibition potential against the CD44 biomarker in breast CSCs [154]. Additionally, HA/chitosan nanoparticles loaded with CM11, a short cationic antimicrobial peptide, showed significantly higher cytotoxicity and ability to induce apoptosis in various cancer cell types, including lung adenocarcinoma, neuroblastoma, and pancreatic carcinoma cell lines, by targeting CD44 [155].

Natural compounds and chemotherapeutic agents can indirectly inhibit CD44 isoforms overexpressed in cancer cells and CSCs. Salinomycin (SLM) alone or combined with paclitaxel, sulfasalazine (SSZ), Zerumbone (ZER), a combination of epigallocatechin gallate (EGCG) and curcumin, silibinin, galangin, and apigenin are some examples. These compounds have all shown promise in reducing the CD44+ CSC population in different cancer cell lines. They also suppress proliferation, migration, and angiogenesis and induce apoptotic and autophagic cell death pathways in tumor cells, inhibiting some molecular pathways such as the STAT3 pathway, cystine-glutamate transporter, and EMT [156,157,158,159,160,161,162].

Researchers have successfully used HA to target CD44-overexpressing cancer cells. A study by Eliaz and Szoka demonstrated the efficacy of HA-modified liposomes in delivering chemotherapeutic agents to CD44-overexpressing cancer cells, indicating its potential as a targeted delivery mechanism [163]. Encapsulating doxorubicin (DOX) in HA-targeted liposomes was highly effective in killing cancer cells expressing high levels of CD44 [163]. In another study, Spadea et al. examined the expression of CD44 isoforms and HA internalization efficacy in human dermal fibroblasts (HDFs) and different cancer cell lines, including CRC. The study revealed a positive correlation between the expression of CD44s and the uptake level of HA. Additionally, the study indicated that CD44s+ HDFs were less effective in the uptake of HA than CD44s+ cancer cells [164]. 

Researchers successfully delivered SN38 (7-ethyl-10-hydroxy-camptothecin) to human gastric tumors using hybrid nanoparticles that targeted CD44 and HER2. They encapsulated SN38 in nanoparticles comprised of a Poly (lactic-co-glycolic) acid (PLGA) core and a lipoid shell modified with HA and anti-HER2/neu peptide mimic (AHNP) [165]. Additionally, the modification of HA facilitated the efficient delivery of curcumin (CUR)/DOX nanoparticles to hepatocellular carcinoma and human non-small cell lung cancer (NSCLC) for treating MDR cells through CD44 receptor-mediated targeted delivery [166].

The siRNAs can cause gene silencing through translation repression. In NSCLC cells, a designed siRNA inhibited CD44 expression, suppressing cell proliferation and colony formation ability [167]. In EGFR wild-type NSCLC cells, CD44 inhibition attenuated cell growth, promoted cell cycle arrest, stimulated cell apoptosis, and enhanced sensitivity to cisplatin [168]. In CD44+ ovarian cancer cells, MDR1 downregulation increased apoptosis and suppressed ovarian cancer growth [169]. The selective targeting of CD44+ CRC cells has been successfully demonstrated by administering anti-KRAS siRNA that is loaded in poly hexamethylene biguanide (PHMB) and a chitosan complex coated with HA [170]. Similarly, in the same cancer, CD44 was targeted through the direct administration of ON-TARGET plus human CD44 siRNA or indirectly by silencing mucin (MUC5AC) gene expression using a small hairpin RNA construct (pSUPER-Retro-shMUC5AC). As a result of these methods, there was a significant decrease in the expression of CD44-related cell migratory and invasion downstream signaling molecules, such as phosphorylated Src kinases, AKT, and integrin-4 [171].

CD44v6 is an attractive target for CAR T cell therapy. Studies found that CD44v6-CAR T cells controlled tumor growth and increased survival in lung and ovarian carcinomas [172]. CD44-CAR T cells were more effective in suppressing tumor growth and increasing survival in CD44+ hepatocellular carcinoma mice [173]. A bispecific molecule, BiTE, was also created to target CD44v6. It was incorporated into an oncolytic helper binary adenovirus (CAdDuo) that encodes an immune checkpoint blocker (PD-L1Ab) and an immunostimulatory cytokine (interleukin [IL]-12) to form CAdTrio. This CD44-CAdTrio allowed HER2-CAR T cells to effectively kill CD44v6+ head and neck carcinoma cells, improving tumor control and survival [174].

## 5. Conclusions

CD44s and CD44v isoforms are overexpressed in CRC and are crucial in enhancing carcinogenic processes. Although the data are conflicting, studies have shown that increased CD44 expression strongly correlates with higher histological tumor grade, advanced clinical tumor stage, shorter survival, and poor prognosis. CD44 represents a promising target for cancer therapy. Targeting CD44 isoforms can potentially reverse malignant behavior and increase cancer cell sensitivity to therapy. Many therapies targeting CD44 are under investigation, including antibodies, peptides, pharmacological and natural inhibitors, HA-modified nanocarriers, siRNA, and CAR T-cell therapy.

Furthermore, the association between CD44 and the molecular pathways involved in tumor proliferation and progression (Ras-MAPK and PI3K/Akt pathways), the ability to act as a co-receptor of various growth factors (i.e., HGF, VEGFR-2), and the potential correlation with TMB and MSI status open new research scenarios for the development of innovative strategies in cancer treatment. Targeting CD44 can promote more robust responses to target therapies and mitigate intrinsic treatment resistance. Further studies are essential to define and standardize the most appropriate method and score for CD44 expression. Prospective studies on larger samples are also needed to confirm the prognostic and predictive role of CD44 and its possible role as a therapeutic target.

## Figures and Tables

**Figure 1 cancers-16-01569-f001:**
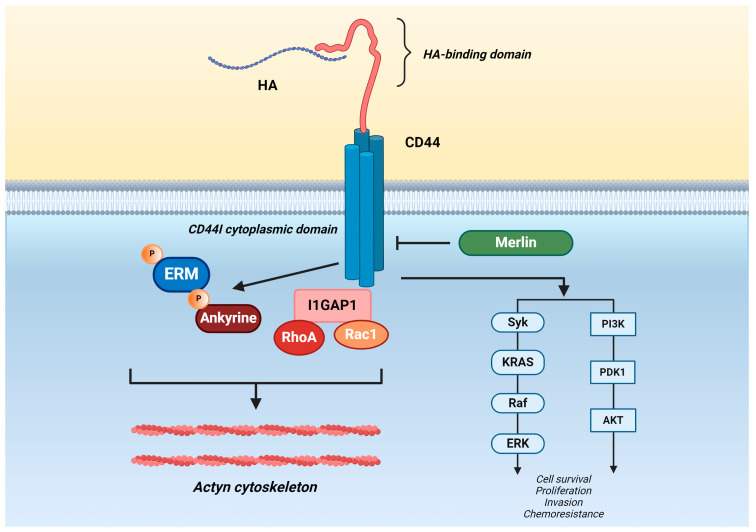
The binding of hyaluronan (HA) to its receptor cluster of differentiation 44 (CD44).

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
