# Peer review of "CD44: A New Prognostic Marker in Colorectal Cancer?"

_cancers, 2024, doi:10.3390/cancers16081569_

Round 1
Reviewer 1 Report
Comments and Suggestions for Authors
The review article "CD44: a new prognostic marker in colorectal cancer?" is primarily focused on compiling the prognostic, predictive and therapeutic role of CD44 in CRC. Overall, the article is well-compiled and well-written with an easier flow of reading. However, I feel that the authors primarily focused on CD44 as a whole or its variants only, and ignored the importance of its structural domains, more specifically the cytoplasmic domain, which is crucial to activate most of the CD44-mediated signaling pathways, and is the key to CD44-specific roles in most of the cancers. Given a limitation of a review article, it reads good, but it can be improved by including a subheading for compiling the role of individual domains of CD44.
Author Response
Thank you for your comments. We appreciate your observations. As you suggested, we created a subheading to describe the role of CD44's three domains.
Reviewer 2 Report
Comments and Suggestions for Authors
it is an interesting subject however there is not any novelty. There are a number or reviews on this topic.
Author Response
R: Thank you for your comment. We tried to give our interpretation of the data already present in the literature.
Round 2
Reviewer 1 Report
Comments and Suggestions for Authors
Thanks for making changes to improve the flow of the reading and the overall importance of this article.
Reviewer 2 Report
Comments and Suggestions for Authors
The review did not add in the scientfic knowdledge. Lack of innovation